# An Update on Viral Hepatitis B and C in Mexico: Advances and Pitfalls in Eradication Strategies

**DOI:** 10.3390/microorganisms12071368

**Published:** 2024-07-03

**Authors:** Marina Campos-Valdez, Manuel Alejandro Castro-García, Martha Eloísa Ramos-Márquez, Carmen Magdalena Gurrola-Díaz, Adriana María Salazar-Montes, Laura Verónica Sánchez-Orozco

**Affiliations:** Instituto de Investigación en Enfermedades Crónico Degenerativas, Centro Universitario de Ciencias de la Salud, Universidad de Guadalajara, Sierra Mojada 950, Independencia Oriente, Puerta 7, Edificio Q Segundo Nivel, Guadalajara 44340, Jalisco, Mexicolnh.macg@gmail.com (M.A.C.-G.);

**Keywords:** viral hepatitis, hepatitis B, hepatitis C, Mexico, epidemiology, prevention, eradication

## Abstract

In Mexico, hepatitis B and C infections are a significant burden on the health system. The aim of this narrative review was to analyze the state of the art on hepatitis B and C in Mexico by searching and studying available data in academic articles and government reports and statements on epidemiology, prevention, treatment, and elimination strategies undertaken by the Mexican government. Even where the government has implemented a hepatitis B vaccination strategy to reduce its incidence, a very low proportion of people complete the vaccination schedule. Regarding hepatitis C, there is a National Elimination Program that emphasizes the importance of screening, diagnosis, and treatment focused on the population at risk. With the implementation of this program, more than a million fast tests have been carried out and the positive cases have been verified by viral load. Infected patients are tested to determine liver function, fibrosis stage, and coinfection with HBV and/or HIV. Patients without cirrhosis and/or coinfections are treated in first-level care centers, while those with cirrhosis and/or comorbidities are referred to specialists. The possibility of hepatitis C eradication in Mexico seems more likely than eradication of hepatitis B; however, major challenges remain to be overcome to reach both infections’ elimination.

## 1. Introduction

Viral infections by hepatitis B virus and hepatitis C virus (HBV and HCV) can cause a wide spectrum of clinical manifestations: asymptomatic carriers (who usually do not know they are infected), acute hepatitis, and chronic hepatitis which can lead to cirrhosis and hepatocellular carcinoma (HCC) [1]. In addition, unlike HCV, HBV can cause fulminant liver failure in less than 1% of infected subjects. Hepatitis B is transmitted through the blood, semen, or other body fluids of an infected person [2]. During acute infection, symptoms can manifest at any time from two to six weeks. These symptoms can include fever, fatigue, appetite loss, nausea, vomiting, abdominal pain, dark urine, light-colored stools, joint pain, and jaundice. It can take decades for symptoms to appear in chronic viral hepatitis [3]. Approximately 95% of immunocompetent adults and a small percentage of infected infants (depending of the age of infection) develop antibodies against the hepatitis B surface antigen (anti-HBs), clear the HBV infection and recover; however, HBV persistence for more than six months leads to chronic infection that may be symptomatic in about 10% to 30% of carriers [4]. 

Hepatitis B surface and e antigens (HBsAg and HBeAg) and their antibodies IgM anti-HBc, total anti-HBc, anti-HBe, and anti-HBs constitute the panel for the HBV infection serologic diagnosis. HBsAg detection is the main serologic marker, which is detectable from one to ten weeks after exposure. Positive detection of IgM anti-HBc, as well as HBsAg and HBeAg, indicates acute hepatitis B. Negative IgM anti-HBc with positive HBsAg and the non-detection of any of the serological markers HBeAg, anti-HBc, anti-HBe or anti-HBs indicates chronic hepatitis B. Additionally, the presence of anti-HBs (>10 IU/mL) indicates immunity from spontaneous resolution of acute hepatitis B infection or HBV vaccination response [5]. Nucleic acid amplification tests (NATs) are mandatory to confirm the presence of HBV infection along with treatment follow-up.

Drugs approved by the United States Food and Drug Administration for the treatment of chronic hepatitis B include standard interferon alfa-2b and peginterferon alfa-2a (PEG-IFN) and six nucleotide/ nucleoside analogs (NAs), lamivudine, adefovir, entecavir, telbivudine, tenofovir, and tenofovir alafenamide. Currently, entecavir and tenofovir (or tenofovir alafenamide) are preferred over other NAs due to the high barrier of resistance and their effectiveness. Nowadays, complete cure is not possible although NA therapy is safe. Thus, the goals of therapy are to prevent the development of cirrhosis, HCC, and liver-related death [6,7]. In acute hepatitis B cases, since more than 95% of immunocompetent adults recover spontaneously, in most cases treatment is not recommended. Treatment is indicated only for patients with fulminant hepatitis and those with prolonged, acute severe hepatitis persisting for >4 weeks [8].

Hepatitis C transmission occurs by the same routes as hepatitis B. For symptomatic subjects with hepatitis C infection, symptoms typically appear two to twelve weeks after exposure. Hepatitis C can be a short-term illness, but it can develop into a life-long chronic infection in 70–85% of people when untreated. Most people with chronic hepatitis C (CHC) have no symptoms at all or experience non-specific symptoms (chronic fatigue and depression). A routine examination of the patient may show elevated ALT enzyme levels either during acute or advanced infection [5,9,10].

Hepatitis C diagnosis is made via a positive hepatitis C virus antibody (anti-HCV) test. These antibodies can be detected four to ten weeks after infection, and do not differentiate between acute, chronic, or past infection. HCV-RNA is detectable as soon as one to three weeks after infection and confirms the diagnosis of active hepatitis C infection. If HCV-RNA remains positive for longer than six months, it indicates CHC. Currently, the available drugs for CHC treatment are PEG-IFN, ribavirin, and direct-acting antivirals (DAAs). A sustained virological response (SVR) characterized by HCV-RNA non-detection after 12 weeks at the end of treatment indicates successful treatment. More than 90% of those HCV infected patients can be cured with appropriate DAA treatment [5].

It is estimated that globally, 328 million people are chronically infected with HBV or HCV (most of them being undiagnosed and untreated). Olaru et al. (2023) roughly calculate a global pooled seroprevalence of 5.8% for HBsAg and 10.3% for anti-HCV [11]. Hepatitis B and C chronic infections are responsible for over 95% of hepatitis deaths (there are over one million deaths by cirrhosis and liver cancer related to viral hepatitis per year).

The World Health Organization (WHO) aims to achieve viral hepatitis elimination by 2030. Its targets are: (a) to reduce new hepatitis B and C infections by 90%, (b) to decrease the mortality by cirrhosis and HCC by 65%, (c) to achieve hepatitis B and C diagnosis in at least 90% of the infected people, and (d) to ensure treatment coverage for at least 80% of patients who fulfill the clinical criteria to be treated [12]. 

In Latin America, there are scarce studies about hepatitis B and C epidemiology. According to Szabo et al. (2012), the prevalence of HCV infection in Latin America ranges from 0.9% to 5.8%, Mexico having the second-highest epidemiological burden (with an estimated 1.6 million infected persons) [13]. A multicenter study of CHC patients in Latin American Centers (between 2014 and 2015) found that patients’ mean age was 58 (53.9% being female), 39.3% had cirrhosis, 41.2% were treatment-naive, 49.8% were treatment-experienced, and 8.9%, were under treatment with DAAs. Further, 71.8% of the patients received concomitant medications with proton-pump inhibitors (20.8% being the most reported); transfusion was the probable route of infection in 46.8%; and 26.4% presented liver-related comorbidities [14].

The HBsAg seroprevalence in Latin America and the Caribbean has decreased from 2.6% (1990) to 1.9% (2015), and further down to 1.8% (2019). Among children younger than 5 years old, it reduced from 1.1% (1990) to 0.1% (2015 and 2019) [15]. The distribution of HBV seroprevalence in Latin America is heterogeneous; most Latin America countries (Mexico, Honduras, Nicaragua, Costa Rica, Panama, Cuba, Paraguay, Uruguay, Chile, Argentina, Peru, and North Colombia) are considered regions with low seroprevalence. There are regions with intermediate seroprevalence (Guatemala, Belize, El Salvador, Honduras, Haiti, Dominican Republic, Puerto Rico, Ecuador, Venezuela, Guyana, Surinam, French Guyana, and South of Brazil) and a few with high seroprevalence (Peru, South Colombia, Northern Bolivia, and Northern Brazil) [16]. The aim of this narrative review was to analyze the state of the art on hepatitis B and C in Mexico by searching and analyzing available data in academic articles and government reports and statements on epidemiology, prevention, treatment, and elimination strategies undertaken by the Mexican government.

## 2. Search Methods

In this narrative review, a search of relevant published articles was undertaken through examination of databases including PubMed and Scopus from January 2005 to March 2024, to identify studies related to the hepatitis B and C viruses’ epidemiology, vaccination, treatment and risk groups from Mexico. The following search terms alone or matched with “and”/“or” were utilized: “hepatitis B”, “Hepatitis C”, “Mexico”, “epidemiology”, “treatment guidelines”, “blood donors”, “HIV coinfection”, “risk groups”, “consensus”, “vaccination”, “Mexican patients”, etc. Official Mexican government websites were consulted to look for official news, epidemiological bulletins, treatment guidelines, and official health programs for prevention, treatment, and eradication strategies for viral hepatitis. Only published papers in English and Spanish were reviewed. The articles’ final selection was based on the content quality, determined by the background, applied methods, sample selection criteria and size, and adequate data presentation, etc. Studies that were not performed in the Mexican population were excluded.

## 3. Hepatitis C in Mexico

The systemic review by Chiquete and Panduro (2007) showed a low prevalence of anti-HCV antibodies [0.37% (95% CI, 0.36–0.38%)], with blood transfusion as the most frequent risk factor of HCV infection. However, they made clear that the available data were from only six Mexican states and that most of the analyzed population were blood donors; therefore, it might not be representative of the general population [17]. 

The most recent Continual National Health and Nutrition Survey (ENSANUT by its Spanish acronym) from 2018, using a two-step immunoassay (GC_37_, Abbott, Germany) with a sensitivity of 99.1%, reported a seroprevalence of [0.38% (95% CI, 0.24–0.59%)] of anti-HCV antibodies in the serum of adults aged 20 and older; of those, 14.9% had detectable viral RNA (representing 46,000 people in Mexico with active infection). The anti-HCV prevalence was 0.23% (95% CI, 0.11–0.48%) in the population aged 20 to 49 years old, and 0.59% (95% CI, 0.34–1.06%) in those aged 50 and older. The prevalence in people aged 20 to 49 was similar during 2012 and 2018, which suggests that the prevalence of HCV has remained stable [18]. Thus, taking into account the Historical Epidemiological Bulletin reports, the changes in incidence may depend on the state (Figure 1) [19]. At present, the prevalence of HCV in Mexico is reported at 1.4% (with a viremia rate between 0.27% and 1%). Appendix A show data from studies about the hepatitis C prevalence in Mexico.

Based on the 2020 Annual Epidemiological Surveillance Report, there has been an average of 2108 new cases of HCV each year (2010–2020), with an incidence rate of 1.06 cases per 100,000 inhabitants. The Mexican states with the highest incidence rates were Baja California Norte, Chihuahua, and Colima [20,21]. Later in 2023, the states with the highest incidence became Baja California Norte, Sinaloa, Sonora, Baja California Sur, and Chihuahua [19]. 

The law all over Mexico states that blood banks must screen potential unpaid donors by clinical evaluation, applying a structured questionnaire of potential risk factors for infectious diseases and serologic discrimination using specific immunoassays for HBsAg and anti-HBc detection for HBV and HCV, respectively, among other blood-borne infectious diseases [22]. Unfortunately, molecular diagnosis of viral hepatitis by PCR or RT-PCR is still not included in the Mexican normativity for the selection of blood donors.

A study in Mexican patients with CHC found that the main transmission route was blood transfusion (64.2%); nevertheless, when patients exposed before 1995 were excluded, the incidence decreased to 4.5% [23]. Among blood donors, a study of the prevalence of viral hepatitis over a 13-year period at national level (2000–2012; 18,617,288 serological tests during this period of time) showed an increase in seropositive cases [8170 (0.69%) to 10,217 (0.57%)] [24]. In addition, screening of occult hepatitis C infection among blood donors in Ciudad de Mexico (November 2015–July 2016) uncovered a prevalence of 3.4%. The infection cases were significantly associated with risky sexual intercourse and acupuncture [25]. In Puebla state, occult hepatitis C in blood donors was reported to be 0.004% in the period from 2012 to 2015 and from 0.005% from 2017 to 2019 [26]. The difference between these results confirms the relevance of blood-donor selection, and following all legally mandatory criteria in Mexico, and the need for molecular diagnosis of blood-borne viral infections in all blood-donor candidates.

Furthermore, in a screening among healthcare workers (considered an at-risk population for HCV transmission), anti-HCV antibody prevalence varies from 0.08% to 1.03% [27,28,29]. From these studies, only one of the positive subjects was positive for HCV-RNA with normal ALT and infected with genotype 2b [27]. The identified risk factors were blood transfusion before 1992, dental procedures, risky sexual behavior, accidental puncture wounds, and surgical interventions [27,28,29].

In 2020, Mexico’s health system established the National Program for Hepatitis C Elimination (NPHCE); the main targets of this elimination strategy are populations at risk, such as people living with human immunodeficiency virus (HIV), intravenous drug users, prisoners, migrants, hemodialysis patients, and people living in high-incidence municipalities. Epidemiologic studies show that these communities are the groups at the highest risk of HCV infection. However, there are not enough data about the prevalence of HCV infection among Mexicans in these groups; they are disproportionally affected by HCV according to several studies, with a prevalence as high as 43.1% among intravenous drug users, and 3.2% among inmates [30]. A meta-analysis by Sedeño-Monge et al. (2021) indicates that the main groups at risk of acquiring hepatitis C in Mexico are prison inmates, drug users, dialysis patients, people with risky behaviors for sexually transmitted infections (STIs), and healthcare workers (prevalence of 3.34%, 84.25%, 10.78%, 1.6%, and 1.36%, respectively) [31].

Taking global and Mexican data together (the latter being rather limited) regarding HCV infection, it can be assumed that these groups remain as the main targets for screening; however, it might be relevant to consider the importance of enhancing the screening among health workers, because they might be a group at risk.

### 3.1. Hepatitis C Treatment

In Mexico, the treatment with the DAA drugs regimen is free of charge for patients fulfilling the inclusion criteria for treatment. Different treatment algorithms are clearly explained in the National Clinical Guide; patients that are candidates for treatment will be given DAAs such as 100 mg glecaprevir and 40 mg pibrentasvir (Maviret, pan-genotypic) or 400 mg sofosbuvir and 100 mg velpatasvir (Epclusa, pan-genotypic). Follow-up is mandatory for all patients during and after the treatment regimen. After 12 weeks of treatment, tests of viral load and hepatic function (ALT and AST) are performed to evaluate SVR. If hepatic function markers remain elevated, medical assessment must be performed for possible associated liver damage. Patients without SVR are referred to a specialist. All patients with persistent risk factors (drug users, men who have sex with men, people who have unprotected sex, etc.) must be tested annually for viral load and liver function [20]. 

### 3.2. National Program of Hepatitis C Elimination

At the start of 2019, Wolpert-Barraza et al. submitted a proposal with 10 imperative actions for a national program to boost the detection, treatment, and follow up of hepatitis C patients. These proposals included: (1) first-level medical care and identification of potentially infected patients, (2) syndemic surveillance, (3) early detection promotion and diagnosis, (4) early treatment, (5) free-of-charge medication, (6) health promotion and disease prevention, (7) research, (8) evaluation, (9) health worker training, and (10) updates to clinical guidelines [32].

In 2020, the Mexican government responded to the WHO call, through the Ministry of Health (MH) and the National Center for HIV and AIDS Prevention and Control (CENSIDA, by its Spanish acronym), responding with a proposal to bring out the NPHCE by releasing the first bulletin related to this program in July, 2020 [33]. The main objective of this program was to ensure the national response toward the elimination of hepatitis C through coordinated actions like promotion, prevention, diagnosis, and treatment across the entire health system, taking into account gender perspective and human rights, and without any discrimination, focusing on public health for all people and communities. The main points to reach this objective are: (a) universal access for diagnosis and treatment; (b) a standardized record program throughout all health system; (c) first-level medical care; (d) strategies focused on the most affected population. To accomplish the previously mentioned actions, the MH and CENSIDA provides basic information about hepatitis C to the general population, such as how it is contracted, symptoms, risk groups, and how to diagnose. In addition to this, on 12 April 2021, CENSIDA started to customize assistance by phone according to people’s requests.

The health and social institutions that have been collaborating with the MH and CENSIDA in this program are the Mexican Social Security Institute, the Institute of Security and Social Services of State’s Workers, the Coordinating Commission of National Health Institutes, High Specialty Hospitals, Mexican Petroleum, the Marine Ministry, the National Mental Health Board, the National Addictions Council, Academic Organizations, and Community and Civil Society Organization Leaders [33,34].

MH in the Historical Epidemiological Bulletin issued people’s evident response to NPHCE, which was manifested by a noteworthy increase in the detection of new cases in 2023 compared with 2015 (Figure 1). Free treatment and detection campaigns were the key element to triggering people’s response.

### 3.3. National Clinical Guidelines for Viral Hepatitis Prevention and Care

The National Clinical Guidelines for Viral Hepatitis Prevention and Care (NCGHPC) was updated in 2023 by public health institution experts (in the areas of epidemiology, laboratory, health promotion, risk communicators, psychology, psychiatry, pharmaceutical surveillance, internal medicine, gastroenterology, and infectiology) and community leaders [20].

The NCGHPC highlights the emphasis on populations at risk such as (a) people using injectable psychoactive, intranasal, and anal substances; (b) men having sex with men; (c) transgender people; (d) people living with HIV; (e) people with renal chronic disorder on dialysis; (f) people presenting liver disease or hepatic function test alteration; (g) pregnant women; (h) inmates; (i) immigrants; (j) people who have received a transfusion, and those with tattoos and piercings (especially before 1996); (k) people with hemophilia; (l) people who have been subjected to invasive medical, dental, or esthetics treatment; and (m) healthcare workers in risk. This guide also recommends all adults to take an anti-HCV detection fast test at least once in their lifetime.

A rapid serologic test is free for anyone living with at least one of these risk factors; in positive cases, confirmatory diagnosis by viral load must be performed, in addition to liver function tests, blood cell-count, and creatinine, HBsAg, anti-HBc, and anti-HIV tests. As well as this, fibrosis assessment should be conducted with non-invasive methods performed with laboratory panels (fibrosis index-4) or AST-to-platelet ratio index (APRI). A fibrotest or liver elastography is recommended if advanced liver fibrosis is suspected. For cases with a minor fibrosis level, treatments are prescribed at first-level medical care. Cases with an advanced fibrosis level (F3–F4) must be transferred to specialist physicians, as well as patients with extrahepatic illnesses (such as autoimmune diseases, neuropsychiatric, cardiometabolic alterations, coinfections, and treatment failure) [20].

## 4. Hepatitis B in Mexico

Data provided by the Historical Epidemiological Bulletin on HBV infection incidence in Mexico by states show heterogeneous incidence changes (comparing 2015 with 2023); but indicate that incidence remained stable (reduction or increment between −0.2 and 0.2) in states such as Zacatecas, Durango, Guanajuato, Queretaro, Mexico, Chihuahua, Veracruz, Hidalgo, San Luis Potosi, Nuevo Leon, Sonora, Sinaloa, and Baja California Norte. The states with decreased incidence were Aguascalientes, Tabasco, Campeche, Morelos, Puebla, Jalisco, Nayarit, and Cuidad de Mexico. Those in which incidence has increased were Guerrero, Tlaxcala, Colima, Chiapas, Oaxaca, Baja California Sur, Tamaulipas, and especially in Michoacan and Quintana Roo. It is feasible that the incidence increase in certain states has been favored because of an improvement in the screening and diagnosis of HBV infection (Figure 2). In 2020, there were 0.8 cases per 100,000 inhabitants; the states with higher incidence were Quintana Roo, Chihuahua, and Tamaulipas; 75.3% were males and the most affected age groups were 25–44 and 50–59 years old [20,21]. It is interesting that Baja California Norte and Sinaloa were among the five states with more incidence for both hepatitis B and C, but hepatitis B virus did not present as high an incidence as hepatitis C (18.23) in Baja California Norte in 2023. Meanwhile, the highest reported hepatitis B incidence was 1.83 in Quintana Roo. Tijuana, a city on the United States border, is located in Baja California Norte, where illicit-drug users (IDUs) have been screened. Despite both viruses sharing the same transmission routes, it is possible that hepatitis C has been transmitted with a higher frequency in IDUs. In addition, hepatitis B infection is removed in approximately 95% of immunocompetent adults; otherwise, hepatitis C infection tends to progress to chronicity in approximately 70% of infected patients. This is probably another reason for the difference in the incidence of both viruses.

Among HBV infected Mexicans, genotype H is the main circulating HBV genotype (more than 70% of infected people); furthermore, drug resistance mutations have been recognized in patients with chronic hepatitis B [35]. The prevalence of hepatitis B in Mexico is low (<2%); only 0.2 to 0.5% of the population is HBsAg seropositive. 

Risk factors (risky sexual behavior, multiple sex partners, men who have sex with men, blood transfusion, tattooing, etc.) associated with infected subjects with hepatitis B match with the ones reported in low prevalence areas. As previously mentioned, these risk factors are also common in Mexican HCV-infected patients. In healthcare workers, the reported HBsAg prevalence was similar to that of the general population [27,28,36]. Nonetheless, an HBsAg higher prevalence of 7.1% (95% CI, 4.4–9.7%), associated with hemodialysis treatment for longer periods and with more blood transfusions, has been reported in patients with renal disease [37]. Similar HBsAg prevalence (7%) was reported in multiple transfused recipients [38]. 

A study by Rojo-Medina and Bello-López reported that there was a decrease in HBV seropositive cases going from a prevalence of 0.47 to 0.15 between 2000 and 2012 [24]. An analysis of the population aged between 10 and 25 years old (from October 2011 to October 2012) revealed a 0.23% (0.10–0.52%) seroprevalence due to natural infection without evidence of acute or chronic infection [39]. Another report analyzing 28,016 blood donors during 2014–2017 showed an average prevalence of 1.05% for anti-HBc positivity; it also pointed out low educational background and age over 50 years old as the main risk factors; additionally, nine cases of occult hepatitis B infections (OBIs) were identified among the anti-HBc-positive donors [40]. In 2018, the HBsAg prevalence in the adult population was 0.51% (0.54% in women and 0.46% in men) [41].

Remarkably, in 2010, Román et al. reported a prevalence of 6% for HBsAg and 33% for anti-HBc in the indigenous population. Similar results were reported for another indigenous population with an HBsAg prevalence of 10.5% and 6.8% for Mixtecos and Purepechas, respectively. Indigenous communities are described as areas of high endemicity, which could increase the number of hepatitis B patients in the country to seven or eight million, and approximately one million active chronic HBV carriers [42,43].

Furthermore, a high prevalence of OBI has been reported in Mexican patients infected with HIV. Recently, it was found that 36% of patients with HIV were OBI carriers (infected with genotype H of HBV) [44]; in contrast, the prevalence of OBI among blood donors in Mexico appears to be low, as well as in kidney-transplant recipients [40,45,46], but it could fluctuate depending on the studied population. It is possible that the OBI cases might be related to HBsAg mutations that may affect the sensitivity of diagnosis methods [47].

### 4.1. Anti-Hepatitis B Vaccination

The WHO has recommended the incorporation of anti-HBV vaccines in health programs since 1991. In 1999, the anti-HBV vaccine became part of Mexico’s immunization regimen. Currently, the indication is universal vaccination, and all persons born before 1999 and all those who for whatever reason were not vaccinated at birth should receive the vaccination regimen [48].

The National Immunization Program guidelines describe the anti-hepatitis B vaccination schedule as follows: first dose before seven days; second dose, two to eleven months; third dose, six to eleven months after birth [49]. Starting at 11 years old, doses depend on the type of vaccine presentation; for those who have not been previously vaccinated, the 10 µg presentation requires three doses (the first at one month, the second at three months, and the third at six months) [39]. For the 20 µg vaccine presentation, two doses should be given (the second shot four weeks after the first one) [50].

ENSANUT 2012 results showed that vaccine-derived immunity (positive ≥ 10.0 IU/mL for anti-HBs and negative to anti-HBc) was 44.7% overall; it was lower in persons aged 20 to 25 years old (40.83%) than in persons aged 10–19 years old (47.7%) [39]. A cross-sectional study at the General Hospital of Acapulco (in the state of Guerrero) found that among 834 workers, only 52% had been vaccinated against hepatitis B once during their working lives, and only 5.5% met the criterion for a complete vaccination schedule [51]. A study by Lopez-Hernández et al. (2023) suggests that most Mexican children are not receiving their full vaccine schedule on time; in the case of immunization against hepatitis B, 61.37% of children under 12 months received the first dose of the HepB.1 vaccine; 32.03% received the second dose HepB-2; and only 4.16% received the third dose on time. This situation shows that the problem of immunization among the population is due to the high dropout rate (children who receive immunizations but do not complete the series which decreases significantly from the second to the third dose) [49].

### 4.2. Hepatitis B Management

Mexico does not have a National Program to eliminate hepatitis B Virus infection. The NCGHPC has recommended screening via HBsAg and anti-HBc detection mainly in high-risk populations (the same ones indicated for HCV infection). This recommendation is also extended to travelers or people born in geographical areas with moderate to high endemicity to HBV, newborns from HBsAg-positive mothers, organ donors and receptors, and people infected with HIV and HCV. In negative HBsAg cases, vaccination is recommended. For HBsAg-positive subjects, tests for HCV and HIV screening and liver disease biomarkers must be performed, to correctly assesses prognosis as well as to identify the patients who may require treatment. As in the hepatitis C clinical guideline, for hepatitis B, algorithms are specified for each group of patients, including special populations such as co-infected patients, subjects with chronic renal disease, etc.; as well as this, the guideline specifies the surveillance and monitoring of patients who are not candidates for treatment. For both hepatitis B and hepatitis C, a mental health assessment and a legal-or-illegal drug use history are recommended. For positive cases, antiviral treatment should be approached in a multidisciplinary manner with the support of mental health specialists [20]. 

### 4.3. Hepatitis B Virus Genotype H

Unfortunately, there are no published clinical trials in Mexican patients related to treatment response evaluation in chronic hepatitis B–infected patients with genotype H; neither are there studies on the natural history of hepatitis B and its associated changes in serological and biochemical markers, fibrosis grade, and HCC association focused on genotype H infection. In addition, there is not enough research related to the identification of prognostic markers of liver damage severity, nor extended studies on the molecular epidemiology in Mexican geographical areas of likely intermediate or high HBV prevalence (regions inhabited by indigenous populations). It is now necessary to form a multidisciplinary team to conduct multicenter studies to answer these and many other questions, and thus contribute to the search for solutions that will provide valuable strategies for hepatitis B elimination in Mexico. 

For treatment-qualified patients, the recommended options are PEG-IFN or NAs with a high resistance barrier, such as entecavir, tenofovir, or tenofovir alafenamide. Meanwhile, lamivudine, adefovir dipivoxil, and telbivudine are currently not recommended due to their high-resistance mutations identified due to long-term treatment [52,53].

Three types of “cure” for chronic hepatitis B have been defined: (1) partial cure (undetectable serum HBV-DNA and normal ALT level maintained after treatment cessation, but with detectable HBsAg); (2) functional cure (sustained undetectable serum HBV-DNA and HBsAg); (3) complete sterilizing cure (undetected serum HBsAg, with complete eradication of HBV-DNA from serum and liver including intrahepatic covalently closed circular DNA (cccDNA) and integrated HBV-DNA fragments) [52,54]. The last one is rarely achieved; therefore, functional cure is the ideal endpoint of antiviral treatment for chronic hepatitis B. IFN promotes the innate and adaptative immune response in addition to inhibiting the transcription of the HBV viral genome; unfortunately, its adverse effects are not tolerated by some patients. Meanwhile, NAs exert their action through the blockade of HBV transcription and are well tolerated [52]. Currently, studies use different treatment strategies such as monotherapy (initial combination, add-on, or switch-to) with a combination of IFN and NAs. Meta-analysis has shown that the initial combination of NAs and IFN was the most common strategy applied. Results show that NAs being switched to IFN therapy, as well as IFN add-on NAs strategy, could improve the outcome of HBsAg loss [54,55].

Although the reported prevalence for HCV is higher than that of HBV, the chances of achieving a significant reduction in hepatitis C are more encouraging compared with hepatitis B. This is primarily because the current treatment with DAAs is finite, and more than 95% of patients who adhere to treatment will be able to eliminate the HCV in a maximum of 12 weeks [52,53]. In contrast, in most cases HBV treatment fails to eradicate cccDNA and to eliminate hepatocytes containing the integrated viral genome. High cccDNA transcription activity is associated with higher HBsAg quantification (qHBsAg). In clinical practice, qHBsAg can be used in association with HBV viral load to classify patients during the natural history of hepatitis B and to monitor therapy using PEG-IFN or NAs [56,57]. However, most studies have been conducted in Europe and Asia where different genotypes predominate (generally comparing A with D and B with C) [58,59]. In addition, it has been observed that there are differences in baseline HBsAg levels depending on genotype, with these being highest for A, followed by B, C, and D. Furthermore, during treatment, the HBsAg kinetics differ according to HBV genotype; for example, the qHBsAg difference between responders and non-responders to PEG-IFN was higher for genotype A in weeks 12 and 24, and for genotypes B and D in week 12, while for genotype C there was no significant difference at any time [58]. Is important to emphasize the absence of studies reporting this type of information regarding the infection with genotype H.

## 5. Advances in the National Program for Hepatitis C Elimination

For HCV, the Mexican Program for the Elimination of this infection has developed strategies to increase the coverture of HCV infection treatment, such as: Education and training through specialized courses (examples are shown in Table 1).Implementation of first-level care units. To date, more than 600 health-care units are operating in the NPHCE.Telementoring, virtual education, and communication strategy is used to create a space for academic interaction between medical professionals of different specialties and first-line medical doctors working in different geographic areas, in order to achieve clinical orientation for a better diagnosis and treatment for hepatitis C. Healthcare workers in first-level or second-level attention have been supported by telementoring programs with clinical experts. During 2022 and 2023, 58 telementoring queries were performed. The treated cases using these online tools were related to hepatitis C: antiviral treatment failure, pregnancy, HIV coinfection, epilepsy, and psychoactive drug use.Integration of health promoters and nurses to apply rapid serological tests and provide pre- and post-test counseling. Trained promoters have been advancing in health services and reaching drug users and people at risk of acquiring HCV in nightclubs and meeting places for substance use. A commitment was established with representatives of nightclubs and community leaders (LGBTTTIQ+ and drug users) to install temporary modules within the selected sites to facilitate timely access to screening services. The implementation of this strategy in several nightclubs and civil associations was accepted by leaders, representatives, and users, obtaining favorable results in a short time. The interest of the users in the information provided by the promoters was decisive for the screening of this population for STIs.Online awareness campaigns are implemented to attract subjects who are suspicious to being infected; direct phone calls were made or email messages were sent to contact infected patients without the opportunity to be treated previously or with interferon treatment failure.The creation of a standardized record program throughout the health system allows health workers in charge to manage all the information related to this program (screening, diagnosis, treatment, patient follow-up, and supply requirements). The storage of the information related to this program is located in a server named Health Care Administration and Management Environment (AAMATES by its Spanish acronym). Authorized personnel feed the database and it is becoming an important source to evaluate the achievements in the NPHCE.

Among others results from the strategies, 2,670,730 rapid tests for anti-HCV have been distributed in the country up to December 2023. From the beginning of the program to 19 July 2023, 1,001,658 rapid tests were performed (2023, vol 3) [60]. Of these tests, 30,118 were positive and viral load was detected in 16,559. In addition, 77% of people living with HIV treated with antiretroviral drugs were screened for HCV. Treatment with DAAs was provided to 70% of patients. At the moment, in the published bulletins from the NPHCE, there is no information about the follow-up and SVR of these patients. Figure 3 shows the risk factors found associated with the infected subjects.

In Mexico, the NPHCE was launched in 2020 and is compatible with international WHO programs. Certainly, some lessons could be learned from the experience of other programs; for example, Georgia, USA, where the world’s first national hepatitis C elimination program was launched in 2015, has had great success in integrating HCV testing and treatment into primary care. An important accomplishment of Georgia’s program was the introduction of HCV core Antigen (HCVcAg) testing, which is faster, easier, and less expensive than HCV-RNA testing. In addition, this program has pioneered mobile units which improved diagnosis efficiency in remote areas [61]. Egypt, a lower middle-income country and with the highest burden of hepatitis C infection worldwide, has implemented an effective hepatitis C elimination program since 2017 thanks to the relevant negotiations of the Egyptian government, which achieved affordable prices for anti-HCV, NAT-HCV, and DAAs treatments. Despite all these efforts, Egypt still faces major challenges to meet the WHO aims related to the HCV elimination program [62]. In Tijuana, a city in Mexico where people who inject drugs (PWIDs) are an important high-risk group for HCV transmission, a dynamic, deterministic model of HCV transmission, disease progression, and harm reduction among current and former PWIDs (90% anti-HCV positive) was constructed. Modeling suggests that elimination targets are achievable in Tijuana through scale-up of harm reduction with high coverage needle/syringe donation programs plus opiate agonist therapy and DAAs treatment [63]. Mexico should be aware of its own experiences and attainments, as well as those of the successful programs around the world, to contribute to this significant WHO agenda by 2030.

## 6. Improvement Opportunities

Despite the support provided for hepatitis C screening and the confirmation by viral load, biochemical liver function tests, fibrosis-level non-invasive methods, and free treatment for Mexican patients, the effort to eradicate hepatitis C by 2030 is indeed an ambitious challenge. Meanwhile, as long as there are positive cases, there will be a possibility of new contagions. Even when a strict questionnaire is made in blood banks to identify risk factors, there could still be transmission risk by blood transfusion. The window period for serologic methods, a mandatory screening test in blood banks, is already a blockage for early case detection in asymptomatic people with this infection. Only a few blood banks in Mexico perform screening via NAT and despite the low seroprevalence in the country, the presence of seronegative and NAT-positive samples, although scarce [64,65], may impact post-transfusion hepatitis infection because several patients are transfused with just one transfusion package in some cases. Implementation of HCVcAg as an additional test for blood donors to reduce the window period could be easier than NAT implementation. There is little research that reports on the follow-up of patients with blood transfusions or organ transplants.

Running mobile units will be a great advantage to covering places at long distances from the primary health care services. 

Official report databases need to inform about the proportion of patients that have been provided with treatment, how many of them completed the treatment scheme, and how many had SVR. Right now, there is information available about the number of HCV-infected patients, but there is no explicit information about the adherence to treatment and SVR.

On the other hand, as reported for the drug user population, drug use behavior in Mexico has been changing. The 1994–2021 analysis shows a decrease in cocaine and heroin consumption and an increase in crystal methamphetamine; the rise of injectable drug use is still a difficult challenge for hepatitis elimination.

For hepatitis C, people who do not respond or do not fulfill the criteria for treatment could be potential HCV transmitters, in addition to the progression of liver disease that burdens the health systems and complicates their quality of life.

In the case of hepatitis B infection, it is important to promote awareness about the vaccination program against this infection and remind the general population that just one shot of the vaccine is not enough to obtain long-term immunity, and to highlight the importance of finishing the scheme once it is started. In addition, it is essential to start the National Program to Eliminate Hepatitis B Virus Infection (NPEHBV) as soon as possible; it is highly recommended to take advantage of the at-risk groups being evaluated for hepatitis C as a way to include the screening tests for HBV, since both viruses share the same infectious routes although with differences in their frequencies. As soon as the NPEHBV begins HBV screening in all subjects with high-risk factors (not only in cases with positive anti-HCV), and clinical management and cost-free treatment, these strategies, in addition to the vaccination program, will contribute to the control of HBV infection transmission. 

## 7. Future Directions

The National plan to reduce viral hepatitis C in Mexico is making significant progress. However, we suggest strengthening the following points and advancing the hepatitis B elimination plan.
Expand coverage to reach the areas farthest from first-level care centers.Implement the diagnosis for OBI in hepatitis C–positive patients, as this could complicate the progression of liver disease. There is scientific evidence demonstrating that after hepatitis C elimination in DAAs-treated coinfected patients, hepatitis B is reactivated and the liver damage continues.Integrate a National Program for Hepatitis B Elimination into the already advanced NPHCE. This will allow the identification of HBV-infected patients in high-risk groups, especially in stages 1 and 2 from step 1 (Figure 4). From stage 3, it is recommended that screening be carried out mainly in communities with high HBV prevalence (indigenous populations).Management of free access to diagnosis and treatment for hepatitis B-infected patients without medical insurance.Another relevant point is to improve education access regarding the importance of HBV vaccination schedules; one of the strategies that could be used to inform both parents and children would be through lectures in elementary and middle schools.Advance research projects related to viral hepatitis B and C in clinical trials using real-world experience with the subjects included in the NPHCE. Address issues such as genetic and epigenetic factors associated with the progression of liver injury, and response to treatments. Search for OBI and comorbidities that are associated with liver damage in these risk groups, among other topics of interest.

## 8. Limitations of This Study

Hepatitis B and C incidence data in this study was analyzed from Mexico’s Historical Epidemiological Bulletin by the General Directorate of Epidemiology (vol 52 from 2015, 2020 and 2023). No serologic test is indicated in epidemiological bulletins, since it is a collection from all reported data throughout the country. Areas with heterogeneity in hepatitis B and C infections are likely to be found in this country; nonetheless, the findings reported in this review might be mistaken, since there could be a deficiency in the report sending by the care centers. States with the lowest incidence reported for both infections such as Tabasco, Queretaro, Campeche, Morelos, Zacatecas, Estado de Mexico, etc. do not have scientific studies, or such studies are very scarce, regarding viral hepatitis. There are regions where this public health problem is not considered. Scientific studies on this topic continue to be concentrated in geographical areas like Ciudad de Mexico, the West of Mexico, Puebla, Veracruz, and Nuevo Leon. Regardless of an increased number of researchers in Mexico interested in viral hepatitis studies, research needs to be strengthened in different geographical areas, with an emphasis on indigenous populations, as there are still many unsolved questions about this health problem.

## Figures and Tables

**Figure 1 microorganisms-12-01368-f001:**
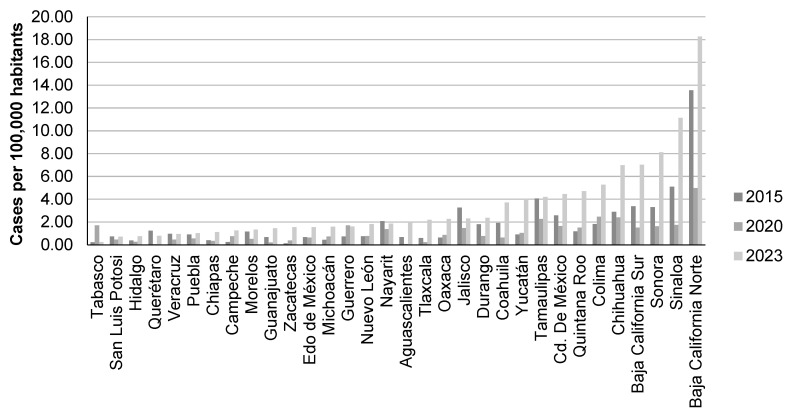
Incidence of HCV infection in Mexico (per state) in 2015, 2020, and 2023. The graph was created using data extracted from Mexico’s Historical Epidemiological Bulletin by the General Directorate of Epidemiology (volume 52 for years 2015, 2020 and 2023) [19].

**Figure 2 microorganisms-12-01368-f002:**
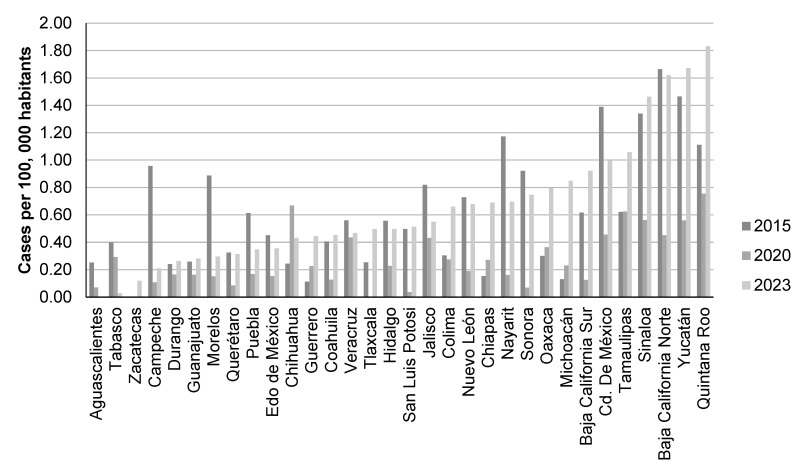
Incidence of HBV infection in Mexico (per state) in the years 2015, 2020, and 2023. Created using data extracted from Mexico’s Historical Epidemiological Bulletin by the General Directorate of Epidemiology (volume 52 for years 2015, 2020 and 2023) [19].

**Figure 3 microorganisms-12-01368-f003:**
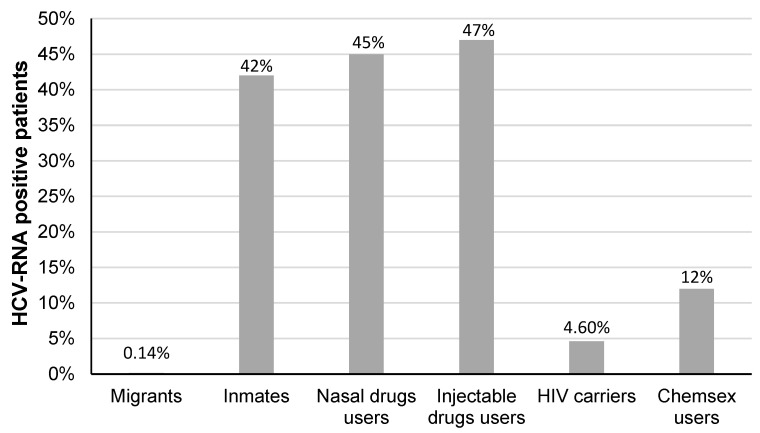
Risk factors identified in HCV carriers [from the beginning of the NPHCE to 27 February 2024 (n = 36,619) registered in the database AAMATES]. (Results were taken from: www.gob.mx/censida/documentos/la-hepatitis-c-es-curable?idiom=es (accessed on 16 April 2024) Vol 5(1), Enero-Marzo., 2024.).

**Figure 4 microorganisms-12-01368-f004:**
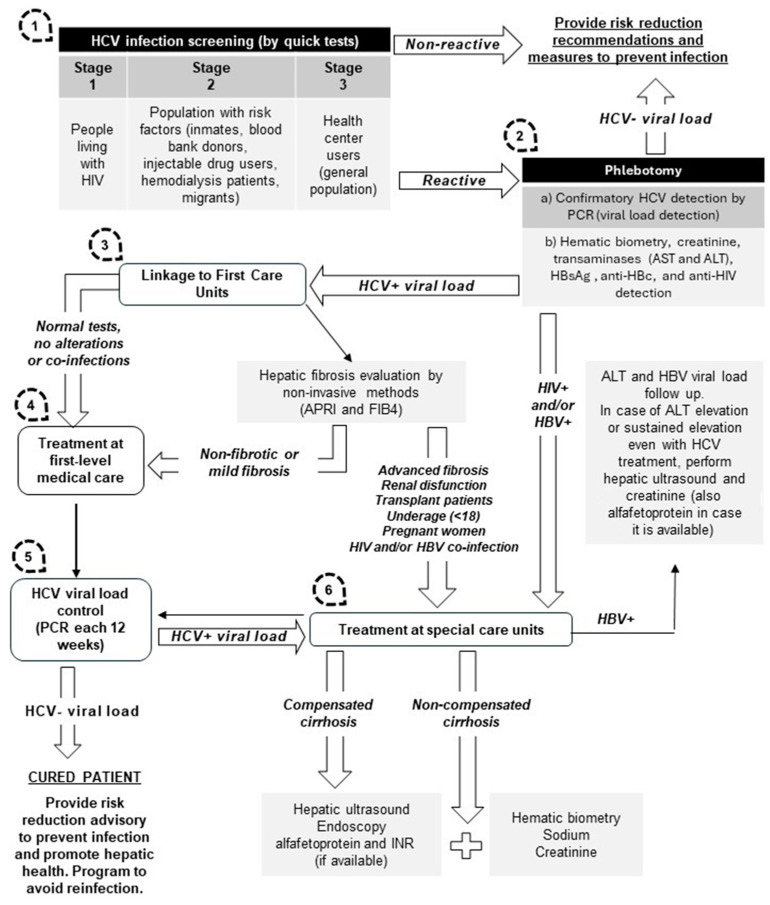
Proposed algorithm on the National Guidelines for Viral Hepatitis Management and followed in the National Program for the Hepatitis C Elimination.

**Table 1 microorganisms-12-01368-t001:** Education and training specialized courses.

Course	2022	2023
Update in the early diagnosis and management of hepatitis C: referral criteria at the first level of primary care	155,647	42,464
Reduction and prevention of harm and risks associated with HIV, HCV, and drug use	16,279	13,782
Access without discrimination to health services for people of sexual diversity	38,543	21,103
Training to PrEP (Exposition Prophylaxis) in Mexico	4530	7902

The number of participants registered in the years 2022 and 2023 is reported in: https://www.gob.mx/cms/uploads/attachment/file/878230/BOLET_N_VHC_CUARTO_TRIMESTRE_2023.pdf (accessed on 16 January 2024).

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
