# Peer review of "An Update on Viral Hepatitis B and C in Mexico: Advances and Pitfalls in Eradication Strategies"

_microorganisms, 2024, doi:10.3390/microorganisms12071368_

Round 1

Reviewer 1 Report

Comments and Suggestions for Authors

This paper provides a comprehensive profile of HCV and HBV infections as well as the suggestions for infections’ elimination. However, this paper has some minor mistakes which need to be modified.

1.     The introduction primarily describes the epidemiological data for hepatitis B and C. It is recommended that the similarities and differences of hepatitis B and C in the context of symptoms, diagnosis and treatment should be introduced more.

2.     Considering the similar transmission channels of HBV and HCV, the risk factors and incidence of HBV infection should be further discussed after describing corresponding information of HCV infection to comprehensively illustrate the important role of controlling the sources of infection and cutting off the channels of transmission.

3.     It is more comprehensive to add some reviews on acute and chronic infections, as well as the corresponding treatment methods.

4. Page 6, lines 234-235 mention the weakness on occult hepatitis B infection (OBI) diagnosis. Please give some advice on improving the screening and diagnosis of OBI.

Author Response

Reviewer 1.

Thank you for your comments which contributed to improve the manuscript.

  1. The introduction primarily describes the epidemiological data for hepatitis B and C. It is recommended that the similarities and differences of hepatitis B and C in the context of symptoms, diagnosis and treatment should be introduced.
  2. A paragraph was added (lines 31-80).
  3. Considering the similar transmission channels of HBV and HCV, the risk factors and incidence of HBV infection should be further discussed after describing corresponding information of HCV infection to comprehensively illustrate the importance role of controlling the sources of infection and cutting off the channels of transmission.
  4. Data about HBV incidence from all states from Mexico was moved to start with the hepatitis B epidemiologic information (lines 274-300). In addition, a paragraph of risk factors for HBV infection was added in page 7; lines 305-313.
  5. It is more comprehensive to add some reviews on acute and chronic infections, as well as the corresponding treatment methods.

R- A paragraph was added in the introduction section. Since treatment guidelines are complex we added a brief piece of information; nevertheless, references were added for more information for the readers. Lines 54-64 and 76 to 80, for hepatitis B and C, respectively.

Finally, English review was made with the assistance of an English edition expert.

Reviewer 2 Report

Comments and Suggestions for Authors

The narrative review concerns epidemiology of HBV and HCV infection in Mexico State. To perform analysis, the authors included available data from academic articles and official reports on the items. The well-known risk factors (e.g., drug usage) are demonstrated for HCV- infected groups. The positive effects of governmental programs in diagnostics, prevention and treatment of HBV and HCV are declared and illustrated, like as outlooks for future. The article may be of some interest, as a part of worldwide studies in this area. However, the results must be compatible with general (e.g., WHO) approach to such extensive epidemiological studies.

Remarks:

The aim of this study should be also highlighted in the Abstract (lines 22-23).

Lines 60-69: Inclusion and exclusion criteria should be presented for the information sources (especially, to avoid duplication of statistical data from various sources). Total numbers of persons (patients) studied, distribution by appropriate risk groups as well as their demographic characteristics (age group, gender, ethnicity, region of living etc.) should be provided in a separate table (or graphs). Distribution of screening data by the time periods (2015-2023) should be also demonstrated.

Line 76 and further…: One should provide the reference values on frequency of HCV-positive tests when screening healthy blood donors, then comparing it with different risk groups. The frequency of HBV positivity in general poopulation should be also given (if possible, by different regions).

Line 95-99. When describing appropriate governmental programs, one should stress possible differences in HBV and HCV diagnostics, and appropriate preventive measures in central and peripheral Mexican states.

In Conclusion, one should mention some limitations of this study, for example: heterogenous coverage of population groups by the still developing diagnostic programs, expecially at first-level care centers. The results may be also influenced by evolution in diagnostic and prevention programs over 2015-2023.

Finally, one should conclude if this study is compatible with international (e.g., WHO) programs of hepatitis screening and prevention

Comments on the Quality of English Language

Only minor copy editing required

Author Response

Reviewer 2.

Thank you for your comments which contributed to improve the manuscript.

The aim of this study should be also highlighted in the abstract.

  1. The aim was added to the abstract. Lines 12 to 15.

Lines 60-69: Inclusion and exclusion criteria should be presented for the information sources (especially, to avoid duplication of statistical data from various sources). Total numbers of persons (patients) studied, distribution by appropriate risk groups as well as their demographic characteristics (age group, gender, ethnicity, region of living etc.) should be provided in a separate table (or graphs). Distribution of screening data by the time periods (2015-2023) should be also demonstrated.

R- Inclusion and exclusion criteria were added in the methodology section lines 126-129. This is a narrative review, we didn´t design this study as a metanalysis review. We added more references and made some tables (supplementary material) to make the information more available for the readers.

Line 76 and further…. One should provide the reference values on frequency of HCV-positive tests when screening healthy blood donors, then comparing it with different risk groups. The frequency of HBV positivity in general population should also given (if possible, by different regions).

R- Reference values were added for the Continual National Health and Nutrition Survey (lines 131-133). Hepatitis B information is scarce. Nevertheless, the incidence cases per state comparing 2015, 2020 and 2023 are shown taking the information from Mexico´s Historical Epidemiological Bulletin by the General Directorate of Epidemiology (Vol 52 from each year). Additional information was added in tables in supplementary material. It was not possible to compare the different populations prevalence’s due to the heterogeneity in the data among the different studies (ages, infection markers, region, populations, etc.)

Line 95-99. When describing appropriate governmental programs, one should stress possible differences in HBV and HCV diagnostics, and appropriate preventive measures in central and peripheral states.

R- Differences in HBV and HCV diagnostics were specified (lines 160-164). In Mexico appropriate preventive measures are mandatory for all states.

In conclusion, one should mention some limitations of this study: for example: heterogenous coverage of populations groups by the still developing diagnostic programs, especially at first-level care centers.  The results may be also influenced by evolution in diagnostic and prevention programs over 2015-2023. Finally, one should conclude if this study is compatible with international (e.g. WHO) programs of hepatitis screening and prevention.

R-Limitations of this study were added lines 570 to 586. About the NPEHC compared with others programs was added a paragraph in lines 481-501.

Finally.  English review was made with the assistance of an English edition expert.

Reviewer 3 Report

Comments and Suggestions for Authors

The authors aim to present a narrative review about viral hepatitis B and C in Mexico.

They largely achieve their goal but they add some paragraphs which are general and informative but they do not include any references relevant to the subject. For example, lines 166 to 170.

I think the paper should be shortened and focus on the review itself instead of trying to be an essay with general conclusions about Mexico' s improvements or future directions. Or change the kind of paper instead of a narrative review.

Comments on the Quality of English Language

English language is generally at a good level. Some corrections are needed, I have added some of my remarks in the text itself, which I have uploaded.

Author Response

Reviewer 3.

Thank you for your comments which contributed to improve the manuscript.

The authors aim to present a narrative review about viral hepatitis B and C in Mexico.

They largely achieve their goal but they add some paragraphs which are general and informative but they do not include any references relevant to the subject. For example, lines 166 to 170.

R: The references were added. The information was added as part of the participation of these institutions in the NPEHC.

I think the paper should be shortened and focus on the review itself instead of trying to be an essay with general conclusions about Mexico' s improvements or future directions. Or change the kind of paper instead of a narrative review.

R-. We try to bring focus on the points we found to be lacking attentions during or data collection while writing this article. An also we try to feed the evidenced found by research, with the governments data and try to constructs and present the most accurate possible “picture” of the situation in Mexico.

Comments on the Quality of English Language

English language is generally at a good level. Some corrections are needed, I have added some of my remarks in the text itself, which I have uploaded.

  1. Thank you very much for your grammar corrections. Finally, English review was made with the assistance of an English edition expert.
